# Balance Impairments in People with Early-Stage Multiple Sclerosis: Boosting the Integration of Instrumented Assessment in Clinical Practice

**DOI:** 10.3390/s22239558

**Published:** 2022-12-06

**Authors:** Ilaria Carpinella, Denise Anastasi, Elisa Gervasoni, Rachele Di Giovanni, Andrea Tacchino, Giampaolo Brichetto, Paolo Confalonieri, Marco Rovaris, Claudio Solaro, Maurizio Ferrarin, Davide Cattaneo

**Affiliations:** 1IRCCS Fondazione Don Carlo Gnocchi Onlus, 20148 Milan, Italy; 2Department of Rehabilitation, Centro di Recupero e Rieducazione Funzionale (CRRF) “Mons. Luigi Novarese”, 13040 Moncrivello, Italy; 3Italian Multiple Sclerosis Foundation, Scientific Research Area, 16126 Genoa, Italy; 4IRCCS Foundation “Carlo Besta” Neurological Institute, 20133 Milan, Italy; 5Department of Physiopathology and Transplants, University of Milan, 20122 Milan, Italy

**Keywords:** multiple sclerosis, balance, inertial sensor, early assessment, preventive rehabilitation

## Abstract

The balance of people with multiple sclerosis (PwMS) is commonly assessed during neurological examinations through clinical Romberg and tandem gait tests that are often not sensitive enough to unravel subtle deficits in early-stage PwMS. Inertial sensors (IMUs) could overcome this drawback. Nevertheless, IMUs are not yet fully integrated into clinical practice due to issues including the difficulty to understand/interpret the big number of parameters provided and the lack of cut-off values to identify possible abnormalities. In an attempt to overcome these limitations, an instrumented modified Romberg test (ImRomberg: standing on foam with eyes closed while wearing an IMU on the trunk) was administered to 81 early-stage PwMS and 38 healthy subjects (HS). To facilitate clinical interpretation, 21 IMU-based parameters were computed and reduced through principal component analysis into two components, sway complexity and sway intensity, descriptive of independent aspects of balance, presenting a clear clinical meaning and significant correlations with at least one clinical scale. Compared to HS, early-stage PwMS showed a 228% reduction in sway complexity and a 63% increase in sway intensity, indicating, respectively, a less automatic (more conscious) balance control and larger and faster trunk movements during upright posture. Cut-off values were derived to identify the presence of balance abnormalities and if these abnormalities are clinically meaningful. By applying these thresholds and integrating the ImRomberg test with the clinical tandem gait test, balance impairments were identified in 58% of PwMS versus the 17% detected by traditional Romberg and tandem gait tests. The higher sensitivity of the proposed approach would allow for the direct identification of early-stage PwMS who could benefit from preventive rehabilitation interventions aimed at slowing MS-related functional decline during neurological examinations and with minimal modifications to the tests commonly performed.

## 1. Introduction

Nearly 2.8 million people worldwide suffer from multiple sclerosis (MS), a chronic demyelinating disease of the central nervous system representing the main nontraumatic cause of disability in young adults [1]. Balance and walking impairments are among the most common deficits in people with MS (PwMS), arising early in the disease course [2,3,4] and gradually progressing over time [5], leading to loss of independence and quality of life [6]. In particular, balance deficits seem to begin earlier than walking dysfunctions, as demonstrated by previous studies showing the presence of static and dynamic balance impairments even in PwMS who walked normally [7,8,9,10,11]. Moreover, static and dynamic balance deficits of early-stage PwMS (expanded disability status scale—EDSS [12]: ≤2.5) have been demonstrated to play a major role in patients’ perception of walking ability during daily life [10,13] and in predicting future falls [10].

Given the strong impact of balance impairments on PwMSs’ disability, several rehabilitation approaches have been proposed with positive results [14,15,16,17]. However, these trainings are usually administered to PwMS who are already in the moderate stages of the disease (i.e., EDSS: 3–6) and, on average, five to fifteen years after diagnosis when symptoms and functional impairments are clearly evident [18]. By contrast, the recent literature highlights the importance of beginning rehabilitation interventions early in the disease course to supplement pharmacological therapy and potentially postpone and/or slow down MS-related functional decline [3,18,19]. In this context, the sensitive detection of subtle impairments, in particular balance deficits, seems a crucial point to planning tailored preventive exercise interventions from the very early stages of the disease [3].

The gold standard for the neurological assessments of PwMS is the EDSS [12], which evaluates seven functional systems (visual, brainstem, pyramidal, cerebellar, sensory, bowel/bladder, and cerebral systems) as well as walking in terms of distance covered without rest or aid [20,21]. EDSS administration includes the evaluation of static and dynamic balance through the Romberg test and tandem gait test, which are part of the cerebellar functional system assessment [21,22]. These tests are considered abnormal if any instability is observed by the neurologist during upright standing with feet together, eyes open and/or closed (Romberg test), or during eight to ten heel-to-toe steps (tandem gait test) [20,23]. Although these tests are parts of the typical neurological examination of PwMS, their results, based on clinical observation, could be not sensitive enough to detect subtle balance deficits in early-stage patients [20,21].

A first possible solution to overcome this limitation could be increasing the difficulty of the tasks already performed, for example, altering proprioceptive inputs by putting a foam pad under the feet during the Romberg test [24]. This sensory condition, in particular, when eyes are closed, has been demonstrated as particularly challenging for PwMS [25] even in the early stages of the disease [8] given the high load on the vestibular system. A second solution, complementing the first one, may consist of using instrumented versions of the clinical tests to obtain more sensitive, quantitative, and objective balance assessments directly in the medical office without the need for larger spaces, longer evaluation time, expensive technologies, or specialized personnel. Wearable inertial measurement units (IMUs) seem to be good candidates to achieve these goals, as demonstrated by the increasing number of studies published in the last fifteen years [26,27] and the growing number of commercially validated devices based on user-friendly technologies (smartphones and tablets) and apps providing objective measures easily and quickly without the need for further data processing [28,29]. Nevertheless, IMUs are still mainly used for research purposes and are not yet fully integrated into clinical practice [30,31]. Published surveys and studies reported that one of the main critical aspects for the routine adoption of these systems is difficulty regarding the time required to understand and interpret the big number of parameters provided [9,30], which are often highly correlated with each other. In this case, the provision of redundant variables does not add further information and makes clinical interpretation unnecessarily complex [32]. Regarding this point, there is a need to reduce the number of instrumented parameters to a few independent variables to increase clinical interpretability, and, at the same time, minimize information loss [33]. Another critical point was underlined by Melillo et al. [9] who highlighted how, apart from a few exceptions [7,8], most of the published literature reported statistically significant differences in balance parameters between healthy subjects and PwMS without providing cut-off values needed to identify balance abnormalities. Moreover, to the best of our knowledge, no studies report cut-off values helping clinicians to understand if the presence of balance abnormalities can be considered clinically significant [31].

Based on the above considerations, in the present multicenter study, we proposed a modified version of the Romberg test (i.e., standing on foam with eyes closed) instrumented with a single IMU on the trunk (ImRomberg). ImRomberg was applied to healthy subjects (HS) and early-stage PwMS with the following aims: (i) to improve the clinical interpretability of the instrumented parameters by reducing the number of data computed from the sensor and by providing cut-off values helping clinicians to understand if an instrumented parameter is abnormal and if this abnormality is clinically significant; (ii) to increase the sensitivity of balance assessment as performed during a typical neurological examination with minimal modifications of the tests commonly performed (i.e., Roberg and tandem gait tests); and (iii) to assess the correlations of the ImRomberg parameters with clinical measures.

## 2. Materials and Methods

### 2.1. Study Design

This cross-sectional multicenter study reports the results of a secondary analysis conducted on the sample of PwMS recruited for a previous work [4].

### 2.2. Participants

A convenience consecutive sample of 81 nondisabled PwMS was recruited from three Italian centers in Milan, Moncrivello (VC), and Genoa. As previously detailed [4], the inclusion criteria were confirmed diagnosis of MS based on McDonald criteria [34], EDSS score ≤2.5, time since diagnosis ≤5 years, stable disease course defined as an increase in EDSS score lower than 1 over the last three months, and age ≥18 years. The exclusion criteria were diagnosis of major depression, severe joint and/or bone disorders interfering with balance and gait (based upon clinical judgment), and cardiovascular or other concomitant neurological diseases. A cohort of 38 healthy subjects (HS) with age and sex distribution comparable to those of PwMS (see Section 3.1) was also recruited as a control group. Inclusion criteria for HS were absence of neurological, cardiovascular, and musculoskeletal diseases and normal balance as measured by a clinical assessment based on the Fullerton Advanced Balance Scale—short version (FABs) [35] detailed in Section 2.3.

### 2.3. Clinical Assessment

Participants underwent the following clinical assessments administered by trained specialized clinicians: Romberg test [36] and tandem gait test [23] (both commonly included in the neurological examination of PwMS [21]), Fullerton advanced balance scale—short (FABs) [35], timed up and go (TUG) test [37], timed 25 foot walk test (T25FWT) [38], and twelve item multiple sclerosis walking scale (MSWS-12) [39,40].

Romberg test. As described by Gill-Body et al. [36], participants were required to stand for 30 s with feet together, arms crossed over their chest, and eyes closed [20,21]. The test was considered abnormal if the person was not able to maintain the position for 30 s or if postural sway larger than that expected by healthy subjects was observed by the clinician. The test was performed with eyes closed only, following previous studies [9,20,21].Tandem gait test. As described by Margolesky and Singer [23], participants were instructed to take ten consecutive heel-to-toe steps along a straight line with eyes open. The test was considered abnormal in case of larger than normal instability causing interruptions during the test.Fullerton advanced balance scale—short (FABs): This scale measures static and dynamic balance during six tasks rated on a five-point (0–4) ordinal scale (maximum score: 24) with higher scores indicating better performances. Scores lower than 23 are considered abnormal [35]. FABs included three items, i.e., Item 1 (“Turn 360° right and left”), Item 4 (“Standing on foam with eyes closed”, here called modified Romberg test, mRomberg), and Item 6 (“Walk with head turns”), which particularly challenge the vestibular system. For this reason, in the present study, these items were referred to as “Vestibular Tests”.A FABs-based balance assessment was used to identify individuals with and without clinically impaired balance: in particular, a person was identified as suffering from clinically impaired balance if at least one of the following conditions applied: (i) FABs total score < 23; (ii) FABs—Item 1 (“Turn 360° left and right”) score < 4; (iii) FABs—Item 6 (“Walk with head turns”) score < 4. Otherwise, the person was identified as having clinically normal balance. Conditions (ii) and (iii) were included to be more inclusive on the identification of persons with balance impairment since FABs—Item 1 and 6 demonstrated to be the most difficult tasks for early-stage PwMS [4].Timed up and go (TUG) test. This test measures mobility and dynamic balance. Subjects were instructed to stand up from a chair, walk 3 m, turn, walk back, and sit down. Time to complete TUG was recorded by the examiner using a stopwatch [37].Timed 25 foot walk test (T25FWT). The T25FWT measures the time taken by the subject to walk for 7.62 m “at their fastest but safest speed” [38]. The test was repeated twice, and the mean duration was computed.Twelve-item multiple sclerosis walking scale (MSWS-12). The MSWS-12 is a self-rated questionnaire on walking ability. The questions focused on the self-perceived impact of MS on 12 balance and locomotor activities of daily living in the last two weeks. In particular, question 4 asks the participant to evaluate how much MS makes it difficult to stand while performing an activity. The transformed total score ranges between 0 and 100 with higher scores indicating higher perceived walking difficulties [39,40].Romberg Test, tandem gait test, and FABs were administered to PwMS and HS while TUG, T25FWT, and MSWS-12 were administered to PwMS only. All tests were performed on a single day in random order.

### 2.4. Instrumented Assessment and Data Processing

Instrumented assessment of standing balance was performed by PwMS and HS during Item 4 of the FABs (instrumented modified Romberg test—ImRomberg). As described by Rose et al. [41], participants were instructed to step up onto a foam pad (Balance-Pad, Airex AG, Switzerland; dimensions: 48 cm × 40 cm × 6 cm, weight: 0.7 kg), fold their arms across their chest, and stand for 30 s with feet shoulder-width apart and eyes closed. The subjects executed the test wearing an inertial sensor (MTw, Xsens Technologies B.V., Enschede, The Netherlands; dimensions: 47 mm × 30 mm × 13 mm, weight: 16 g) secured at sternum level with an elastic band. The sensor consisted of a three-dimensional accelerometer (±160 m/s^2^ range), a three-dimensional gyroscope, (±1200 deg/s range), and a three-dimensional magnetometer (±1.5 Gauss). Signals from the inertial sensor were acquired with a sampling frequency of 75 Hz. Although the sensor is usually positioned on lower back in most of the published literature [26], in the present study, we chose to place it on the sternum, as already done in previous works [42,43]. This choice was ascribed to a preliminary analysis on 33 HS and 36 early-stage PwMS who contemporarily wore lower back and sternum sensors. The results indicated that the between-group effect sizes averaged over all parameters were approximately 39% larger using the sensor on the sternum compared to the one on the lower trunk. This, in turn, indicated a higher discriminant ability between HS and PwMS by placing the sensor in the former position.

Trunk anteroposterior (AP), mediolateral (ML), and vertical (VT) accelerations were reoriented to a horizontal–vertical coordinate system [44], and the middle 20 s of the signals were processed to compute a set of 21 instrumented variables (see Table 1) commonly used to assess standing balance in PwMS [7,8,25,45,46,47,48,49,50].

Data processing was performed using MATLAB R2017b (The MathWorks, Inc., Natick, MA, USA).

### 2.5. Statistical Analysis

The sample size was computed based on our preliminary analysis, revealing significant differences between HS and early-stage PwMS in several instrumented variables descriptive of balance [58]. To be more conservative, considering the variable ML sample entropy, which showed the lowest effect size (HS mean ± SD: 1.7 ± 0.3; PwMS: 1.4 ± 0.5; Cohen’s *d*: 0.70) [58], we found that 105 participants (35 HS and 70 PwMS) were necessary to obtain a difference between groups given N_HS_/N_PwMS_ = 0.5; α = 0.05 and 1−β = 0.9.

PwMS and HS were compared by means of chi-square test (χ^2^) for sex distribution and Mann–Whitney U test (MWt) for all the other clinical and instrumented variables. Benjamini–Hochberg (B–H) correction for multiple comparisons (false discovery rate of 5%) was applied when necessary [59].

Regarding the 21 instrumented variables computed from the ImRomberg test and described above (see Section 2.4), only those revealing a statistically significant difference between HS and PwMS were further analyzed. In particular, these variables (computed on the whole sample including both HS and PwMS, *N* = 119) were first standardized (by subtracting the mean and dividing by the standard deviation) and then entered into principal component analysis (PCA) with *varimax* orthogonal rotation in order to reduce data dimensionality and obtain a smaller set of uncorrelated “latent” variables (i.e., principal components; PCs) that are linear combinations of the original standardized variables [33]. PCs were retained if associated eigenvalues were larger than 1 and if the percentage of the total explained variance was at least 75% [60]. For each PC, factor loadings were considered significant if larger than 0.60 [33].

The appropriateness of PCA was firstly analyzed by inspecting the Pearson’s correlations matrix among instrumented variables: Only those variables showing a correlation coefficient larger than 0.3 and lower than 0.9 (absolute value) were included in PCA to guarantee an adequate correlation and, at the same time, avoid multicollinearity [61]. The suitability of PCA was investigated using the Bartlett’s test, assessing sphericity, and the Kaiser–Meyer–Olkin (KMO) test, assessing sampling adequacy. The PCA was considered viable if Bartlett’s test was significant at *p* < 0.05 and if KMO values, related to the overall variables and the single variables, were greater than 0.50 [62]. Eight outliers were identified in the dataset (Mahalanobis distance, *p* < 0.001 [33]), hence, PCA was rerun after their exclusion. Since the results did not change, the solution obtained on the whole sample was considered in the following sections.

PCA was considered appropriate if the above conditions were met. In this case, the resulting PCs scores were compared between HS and PwMS by using Mann–Whitney U test with Benjamini–Hochberg (B–H) correction for multiple comparisons. For each PC, the area under the receiver operating characteristic curve (AUC) was computed to assess its discriminant ability. Moreover, for each PC, a normative cut-off value was defined as the 95th (or the 5th) percentile of HS data depending on if its increase (or decrease) was indicative of poorer balance.

The same analysis was performed to compare PCs scores between participants with or without clinically abnormal balance. Again, a cut-off threshold indicating a clinically abnormal value was defined for each PC as the 95th (or the 5th) percentile of data related to individuals with clinically normal balance (see Section 2.3 for the definition of clinically normal and abnormal balance).

Chi-square test was used to compare the number of PwMS showing abnormal scores of ImRomberg PCs with the number of participants showing abnormal clinical scores in Romberg and mRomberg tests.

Finally, the association between instrumented PCs scores and clinical scores was performed using Spearman’s correlation coefficient (r_s_). Absolute values of r_s_ between 0.20 and 0.40 indicate small correlation, between 0.40 and 0.60 moderate correlation, between 0.60 and 0.80 strong correlation, and between 0.80 and 1 very strong correlation [63].

Statistical analysis was conducted using SPSS v.20 (IBM, Armonk, NY, USA).

## 3. Results

### 3.1. Sample Description

Demographic and clinical characteristics of HS and PwMS are reported in Table 2. PwMS and HS showed comparable age and sex distribution. The two balance tests usually administered during neurological examinations of PwMS, i.e., Romberg and tandem tests, were normal in all HS and abnormal in 1 (1.2%) and 13 (16%) PwMS, respectively. Similarly, the clinical score of the mRomberg test (FABs–Item 4, standing on foam with eyes closed) was normal in all HS and abnormal in 5 (6%) PwMS. FABs scores were significantly lower in PwMS compared to HS. Forty-five (56%) PwMS reported at least minimally perceived walking limitations during daily living (MSWS-12 score > 0). In particular, 29 (36%) PwMS indicated that MS made it difficult to stand while performing daily-life activities (MSWS-12 Item 4 score > 1).

The 21 wearable-sensor-based variables computed during the instrumented modified Romberg test (ImRomberg) are shown in Table 3 for HS and PwMS. AP and ML sway amplitude, range, velocity, path, total spectral power, and 95% confidence ellipse area were significantly higher in PwMS compared to HS. AP and ML normalized jerk and sample entropy were lower in PwMS, indicating smoother and more regular sway with respect to HS. No statistically significant difference between groups was found in AP and ML 95% power frequency, centroidal frequency, or frequency dispersion, which were, therefore, excluded from subsequent analyses.

### 3.2. Principal Component Analysis (PCA)

From the 15 instrumented parameters showing a statistically significant difference between HS and PwMS, seven (i.e., AP and ML sway range, path, total spectral power, and 95% confidence ellipse area) were excluded since they had Pearson’s correlation coefficients ≥0.90 with at least another variable (Appendix A, Figure A1). By contrast, AP and ML sway amplitude, sway velocity, normalized jerk, and sample entropy was entered into the PCA, given that the absolute values of their correlation coefficients (range: 0.31–0.86) were always between 0.3 and 0.9 (Appendix A, Figure A1).

The Bartlett’s and KMO tests supported the suitability of the PCA. In particular, Bartlett’s test was statistically significant [χ^2^(28) = 897, *p* < 0.001], and KMO overall (0.848) and individual values (between 0.783 and 0.939) were all above the 0.5 threshold. The PCA resulted in two independent PCs (PC1 and PC2; eigenvalues 5.3 and 1.2, respectively) which accounted for 81.9% of the total variance. According to the factor loadings, PC1 was associated with the variables describing jerkiness and irregularity (i.e., complexity) of trunk sway (AP and ML normalized jerk and sample entropy) whereas PC2 was correlated to the parameters descriptive of AP and ML sway amplitude and velocity (here, considered descriptors of sway intensity). Based on these results, PC1 and PC2 were labeled, respectively, as sway complexity (explaining 42.5% of total variance) and sway intensity (39.4% of total variance) (see Figure 1).

Further results derived from the PCA showed a high level of communality, as defined by McCallum et al. [64]. In particular, the resulting communalities (between 0.76 and 0.90) were all above 0.60 and had a mean value (0.82) greater than 0.7. Based on the guidelines of Mundfrom et al. [65], this result (i.e., high communality) indicated that the present sample size (*N* = 119) was adequate for conducting a PCA with a good level of agreement between the sample and the population solution given a two-component model and a variable-to-component ratio of 4 (8:2).

The procedure for the computation of sway complexity and sway intensity scores from the instrumented variables descriptive of ImRomberg is reported in Appendix B, in particular in Table A1.

### 3.3. Sway Complexity and Sway Intensity: HS versus PwMS

As reported in Table 4, PwMS showed lower scores of sway complexity (*p*_B-H_ = 0.003) and higher scores of sway intensity (*p*_B-H_ < 0.001) compared to HS. 

The AUC mean (95% confidence interval) was 0.68 (0.58; 0.78) for sway complexity and 0.72 (0.62; 0.81) for sway intensity, suggesting an acceptable discriminant ability between the HS and early-stage PwMS based on the guidelines proposed by Yang and Berdine [66]. Normative cut-off scores for sway complexity and sway intensity were defined, respectively, as the fifth (i.e., −0.82) and ninety-fifth percentiles (i.e., 0.11) of HS data (see Table 4). By applying these thresholds, specificity was equal to 97% for both components while sensitivity was 31% for sway complexity and 33% for sway intensity, meaning that 25 (31%) of PwMS had an abnormal score in sway complexity (<−0.82) and 27 (33%) had an abnormal score in sway intensity (>0.11).

### 3.4. Sway Complexity and Sway Intensity: Clinically Normal Balance versus Clinically Abnormal Balance Group

Sixty-three participants showed clinically normal balance as defined by the FABs-based assessment described in Section 2.3. This group was composed of all 38 (100%) HS and 25 (31%) PwMS. By contrast, 56 individuals revealed clinically abnormal balance. This group was composed of PwMS only [56 (69%)]. As shown in Table 5, the group with clinically abnormal balance was characterized by lower sway complexity (*p*_B-H_ = 0.002) and higher sway intensity (*p*_B-H_ = 0.041) with respect to the group with clinically normal balance.

The AUC values were 0.68 (0.58; 0.77) for sway complexity and 0.61 (0.51; 0.73) for sway intensity. Clinically significant cut-off scores for sway complexity and sway intensity were identified, respectively, as the fifth (i.e., <−1.01) and ninety-fifth percentiles (>0.59) of scores related to individuals with clinically normal balance. Based on these thresholds, sway complexity showed a sensitivity of 29% and a specificity of 95% while sway intensity had a sensitivity of 23% and a specificity of 95%.

### 3.5. ImRomberg versus mRomberg versus Romberg Test

By applying the thresholds defined above and reported in Figure 2 (i.e., normative cut-offs and clinically significant cut-offs), we found that 26 (32%) PwMS (red circles in Figure 2) showed at least one clinically abnormal instrumented score (i.e., sway complexity and/or sway intensity outside clinically significant cut-offs, the light red area in Figure 2).

Sixteen (20%) PwMS (yellow circles in Figure 2) showed at least one abnormal instrumented score even if not clinically significant (sway complexity and/or sway intensity outside normative cut-offs but inside clinically significant cut-offs, the light yellow area in Figure 2) while 39 (48%) (green circles in Figure 2) showed normal ImRomberg tests (sway complexity and sway intensity inside normative cut-offs, the light green area in Figure 2).

In total, 42 (52%) PwMS (red and yellow circles in Figure 2) presented with abnormal ImRomberg (i.e., sway complexity and/or sway intensity outside normative cut-offs). This percentage was significantly larger than those representing individuals showing abnormal clinical mRomberg [5/81 (6%), *p*_χ2_ < 0.001] and Romberg tests [1/81 (1%), *p*_χ2_ < 0.001] (see Figure 3a).

Thirteen (16%) PwMS showed abnormal tandem gait clinical tests (Figure 3b,c and Table 2). In addition to these subjects, the administration of the traditional clinical Romberg test detected one further abnormal value (1%) (Figure 3b) while the administration of ImRomberg found another 34 (42%) abnormal scores with 21 (26%) of them being clinically significant (two-headed dashed arrow in Figure 3c). In total, clinical tandem and Romberg tests detected fewer (*p*_χ2_ < 0.001) abnormal values [14/81 (17%)] (two-headed solid arrow in Figure 3b) compared to clinical tandem and ImRomberg tests [47/81 (58%)] (two-headed solid arrow in Figure 3c).

### 3.6. Sway Complexity and Sway Intensity: Correlations with Clinical Scales

As reported in Figure 4, the sway complexity of PwMS significantly correlated with TUG, FABs, T25FWT, MSWS-12, MSWS-12 Item 4, and EDSS (|r_s_| ≥ 0.23, *p* ≤ 0.041). A trend toward statistically significant correlation (r_s_ = −0.20, *p* = 0.092) was found with the number of clinically abnormal vestibular tests. Sway intensity correlated only with the T25FWT (r_s_ = 0.27, *p* = 0.013).

## 4. Discussion

In the present study, a modified Romberg test instrumented with a single IMU on the trunk (ImRomberg) was administered to a cohort of HS and early-stage PwMS. The modification of the Romberg test (i.e., standing with eyes closed on foam rather than on a rigid surface) was proposed since the exclusion of vision and the alteration of proprioception increase the reliance on the vestibular system, which is often impaired even in early-stage PwMS, thus augmenting the difficulty of the task [8,24,67,68]. The main goals of the study were (i) to improve the clinical interpretability of the instrumented parameters by reducing the number of data computed from the sensor and by providing cut-off values to help clinicians understand if an instrumented parameter is abnormal and if this abnormality is clinically significant, (ii) to improve the sensitivity of the balance assessment performed during a typical neurological examination with minimal modifications to the tests commonly performed (i.e., Romberg and tandem gait tests), and (iii) to assess the correlations of the ImRomberg parameters with clinical scales. This would allow for direct detection during the neurological examination of those early-stage PwMS who should undergo early balance rehabilitation [18,19].

The analysis of the 21 IMU-based parameters computed from the ImRomberg test revealed that 15 of them were statistically different between HS and early-stage PwMS (see Table 3). However, some correlation coefficients among these measures were particularly high (see Figure A1), indicating a strong (r ≥ 0.80) association between them. This result suggests that these measures probably provide the same information and are therefore redundant. Although a big number of instrumented measures may provide a thorough and detailed characterization of the standing balance, their clinical interpretation may be difficult and time consuming [30,31] due to the presence of redundant, thus, unnecessary, parameters. To partially overcome this problem, principal component analysis (PCA) was used to summarize most of the information (variance) contained in the input parameters into a smaller set of uncorrelated “latent” variables (principal components; PCs) [33] in order to reduce the amount of data to be analyzed and to facilitate clinical interpretation. From the initial dataset of 21 instrumented parameters, eight (i.e., sway amplitude, sway velocity, normalized jerk, and sample entropy in AP and ML directions) were retained and entered into the PCA, which resulted in two principal components (i.e., sway complexity and sway intensity) descriptive of two independent aspects of the ImRomberg test with a total explained variance of 81.9%.

Sway complexity, explaining 42.5% of the total variance, resulted in the first principal component. Looking at the factor loadings reported in Figure 1, the parameters mostly contributing to the sway complexity component were AP and ML normalized jerk and sample entropy, which measure, respectively, the jerkiness and the irregularity (unpredictability) of trunk acceleration during standing. Based on previous studies, sway complexity depends on the number of components involved in the balance control system and on their interaction [69,70]. Healthy balance control is extremely complex since several structures, including visual, proprioceptive, and vestibular sensory networks, central sensorimotor areas, and the musculoskeletal system, simultaneously interact to maintain and adapt standing balance through continuous, automatic, fast, and unpredictable postural adjustments [69,70,71,72]. These, in turn, result in jerky and irregular (i.e., complex) trunk sway. By contrast, low sway complexity, i.e., increased sway smoothness and regularity, reflects the loss of complexity and adaptability of the balance control system due to the reduction and/or impairments of its structural components and their interaction [25,73]. This, in turn, could lead to less automatic and more conscious postural control, as proposed by previous literature [74,75]. From a clinical point of view, lower values of sway complexity are, therefore, indicative of an inefficient control of balance, possibly requiring an increased attentional investment. In line with these concepts, the present results showed that sway complexity computed from the ImRomberg test was significantly reduced in the tested cohort of early-stage PwMS compared to HS, with 25 (31%) subjects showing values below the normative cut-off (<−0.82). A reduction of sway complexity, as measured by the jerk or entropy parameters, has been already documented in PwMS in more advanced stages of the disease (median EDSS ≥ 3) during standing under different perceptive contexts, i.e., eyes open/closed on the rigid surface [7,25,47,49,76] and foam [25]. As previously proposed [25], it can be speculated that the widespread demyelinating lesions due to MS might reduce the connectivity and efficiency of neural subsystems controlling balance [77], leading to the need for a more conscious control of body posture which results in a less complex, smoother, and more regular trunk sway. A novel finding from the present study is that these anomalies, well documented in moderate–severe MS, begin early in the course of the disease and are detectable, at least in the most difficult perceptual condition (i.e., standing on foam with eyes closed), where the only sensory input available is the vestibular one, often compromised also in early-stage PwMS [78,79].

The second component resulting from the PCA and explaining 39.4% of the total variance was labeled as sway intensity. The inspection of factor loadings reported in Figure 1 indicated that the parameters which mostly contributed to this component were AP and ML sway amplitude and velocity. The comparison between the present sample of PwMS and HS showed a statistically significant difference between the two groups, with PwMS showing a larger sway intensity, meaning an increased amplitude and velocity of trunk sway in both the ML and AP directions during the ImRomberg test. In particular, 27 (33%) subjects showed values above the normative cut-off (>0.11). From a clinical point of view, larger and faster trunk movements during standing indicate the difficulty for PwMS in maintaining a stable position at their center of mass far from the limits of the base of support, thus increasing the risk of falls [80]. This result was in accordance with previous studies analyzing standing balance in PwMS in the lower range of the EDSS [7,8,9,21,48,67].

A novel aspect of the present study was the provision of not only normative cut-off values for sway complexity and sway intensity but also clinically significant cut-offs which could help clinicians to detect if there is an anomaly in balance performance and if this anomaly could be considered meaningful from a clinical point of view. By applying these thresholds, we found that 42 (52%) PwMS showed at least one abnormal component (i.e., sway complexity and/or sway intensity) of the ImRomberg test (see Figure 2 and Figure 3a). In particular, 26 (32%) and 16 (20%) PwMS showed abnormalities in the ImRomberg test that were, respectively, clinically meaningful and not clinically meaningful (Figure 3a). These percentages were all significantly larger than those representing the abnormal clinical scores found in the modified Romberg (6%) and traditional Romberg test (1%), thus demonstrating a higher sensitivity of the ImRomberg test compared to its clinical counterparts.

The sensitivity in detecting balance impairment in early-stage PwMS was further increased when the ImRomberg test was combined with the tandem gait clinical test, which is the second balance assessment commonly administered during a typical neurological examination. In fact, the replacement of the clinical Romberg with the ImRomberg test in addition to the tandem gait test increased the percentage of detected balance disturbances from 17% to 58% of cases with minimal modifications of the assessments typically performed during a visit with a neurologist. In our opinion, the current findings may have high clinical relevance. By adopting the proposed approach during routine neurological examinations of early-stage PwMS, about 60% of them could be already advised to undergo early rehabilitation treatments for improving balance as recently suggested [18,19]. However, it must be highlighted that some PwMS showing normal ImRomberg and tandem gait tests could present with dynamic balance deficits that can be detected only through more difficult tasks (i.e., walking over/around an obstacle, walking with head rotations, stairway walking, etc. [11,20,35,81,82]) which are usually executed with physiotherapists in rehabilitation gyms. For this reason, it would be important to advise these individuals, especially if sedentary, to carry out regular physical activity, including dynamic balance.

Regarding the correlation with clinical measures, both sway complexity and sway intensity computed from the ImRomberg test significantly correlated with the T25FWT, confirming that static balance, together with dynamic balance and muscle strength, significantly contributes to gait speed in PwMS [83]. Apart from this significant association, sway intensity did not correlate with any other clinical score. This seems in contrast with previous studies finding statistically significant small-to-moderate correlations between measures of sway amplitude and velocity (i.e., sway intensity) and EDSS [84], MSWS-12 [7,8], and TUG [48]. However, these studies analyzed mixed-MS populations, including participants with an EDSS ranging between 0 and 6.5 and median values (≥2) larger than those characterizing our samples (i.e., 1.5). This leads to the hypothesis that sway intensity becomes significantly associated with the above clinical measures only when considering a sample that includes more severe PwMS. By contrast, sway complexity correlated with all clinical scores, suggesting that this less traditional instrumented measure, proposed as a descriptor of the amount of conscious attention devoted to balancing maintenance [74,75], is more valid and adequate than the classical sway intensity parameters for assessing standing balance under altered perceptive contexts in early-stage PwMS. In particular, sway complexity showed statistically significant associations with dynamic balance scales, i.e., FABs (r_s_ = 0.23) and TUG (r_s_ = −0.29), and a trend towards a statistically significant relationship (r_s_ = −0.20, *p* = 0.092) with the number of clinically abnormal vestibular tests (i.e., standing on foam with eyes closed, turning 360°, and walking with head turns). Despite their statistical significance, these correlations were small, confirming that static and dynamic balance pose different demands to the sensorimotor control system, and, specifically, to the vestibular sensory networks [48,85]. This, therefore, enforced the fact that to obtain a complete balance evaluation, the static and dynamic components should be assessed separately using dedicated tests [11,48]. A significant correlation was also found between sway complexity and MSWS-12, complementing previous results demonstrating a relationship between patient-perceived walking limitations and standing balance under altered visual or proprioceptive inputs [7,8]. Again, the correlation was small (r_s_ = −0.29), in line with previous findings indicating that the major contributors to MSWS-12 score in early-stage PwMS are fatigue, dynamic balance, gait asymmetry, and gait instability [13]. Interestingly, a small correlation was found between sway complexity and Item 4 of the MSWS-12, suggesting that higher perceived difficulty to stand while performing daily-life activities (MSWS-12 Item 4) is associated, at least minimally, to lower sway complexity during the ImRomberg test, i.e., to larger conscious attention in maintaining balance on foam with eyes closed. This finding provided hints for rehabilitation in early-stage PwMS, suggesting that standing balance exercises executed under altered sensory conditions should be included in preventive early rehabilitation treatments together with tasks aimed at improving fatigue, gait symmetry, and dynamic balance, as previously indicated [13]. Finally, sway complexity weakly correlated with EDSS (r_s_ = −0.24). This finding seems in contrast with previous studies demonstrating moderate-to-strong correlations (r_s_: 0.55–0.69) between postural parameters and EDSS during standing with eyes closed on rigid [84] or compliant surfaces [20]. However, these studies analyzed cohorts with a wider range of EDSS levels (EDSS: 1–4.5 [20]; EDSS: 0–6.5 [84]). Moreover, in line with our results, Kalron et al. [84] found no difference in posturographic parameters between subgroups of PwMS in the low EDSS range (i.e., 0–2.5) while they observed a significant progressive worsening of balance starting from the mild–moderate (EDSS: 3–4) to severe (EDSS: 6–6.6) stages of the disease. This, in turn, suggested that a significant correlation between balance performances and EDSS exists only in PwMS with EDSS ≥ 3.

Some limitations must be acknowledged regarding the present study. First, the instrumented assessment was performed only during standing with eyes closed on the foam. It was therefore not possible to compare the sensitivity of this task with that of other instrumented tests, in particular, the traditional Romberg test (i.e., standing with eyes closed on a rigid surface). However, as highlighted by Halmágyi and Curthoys [24], testing an upright stance on a compliant rather than firm surface represents an added value since it provides information about the vestibular function (at least during static conditions) that plays an important role for balance maintenance in early-stage PwMS. Future studies should assess if the present procedure is valid also during the traditional Romberg test. Second, the parameters computed from the raw data recorded by the IMU require signal-processing skills that clinicians do not always have since they are not needed for their role [31]. This drawback could be overcome by relying on recent technologies based on portable devices which already allow for the acquisition/processing of data during clinical tests and automatically provide results just after recording is stopped [29]. In this context, embedding the whole procedure proposed here in a dedicated app and running it on a smartphone could automatically provide the values of sway complexity and sway intensity immediately after test execution without the need for further data processing. In particular, the provision of results through a clear and simple graphical representation reporting the cut-off values here defined would allow a fast preliminary characterization of the patient’s performance by the clinician. Furthermore, such a system would also open up the possibility for the patient to carry out balance assessments autonomously at home and automatically send the results to the clinician remotely. Future studies should analyze this interesting opportunity. Third, the IMU was not used during the tandem gait test. The instrumentation of this test could also further increase the sensitivity of balance assessment performed during standard neurological examination. Fourth, the body heights and weights of the participants were not measured, hence it was not possible to check eventual differences between groups according to these anthropometric parameters which are known to impact balance control. Finally, the number of healthy participants was small and the tested cohort of PwMS consisted of early-stage, nondisabled individuals, thereby reducing the generalizability of present results. Future studies including a larger number of control subjects and more severe PwMS should be performed to provide more robust normative cut-offs, analyze the significance and added value of the above indexes in a more severe population, and examine the test–retest reliability of the method.

## 5. Conclusions

A set of 21 instrumented parameters computed from trunk accelerations during the ImRomberg test (i.e., standing on foam with eyes closed wearing an IMU on the trunk) was reduced with minimal information loss (total explained variance: 81.9%) to two uncorrelated components, i.e., sway complexity and sway intensity, descriptive of independent aspects of standing balance. The significant reduction of the instrumented features (from 21 parameters to two components), clear clinical meaning of the two components, and their association with clinical measures could contribute to facilitating and speeding up clinical interpretation. Sway complexity, descriptive of the automaticity of postural control, was significantly reduced in early-stage PwMS compared to HS, suggesting a less automatic and more conscious control of balance. Sway intensity, related to the amplitude and velocity of trunk movements, was larger in PwMS demonstrating difficulty in maintaining a stable position during the test. To further increase the clinical usefulness of this approach, cut-off thresholds were identified to help clinicians understand if the values of sway complexity and intensity related to a single patient are abnormal and if such abnormality, when present, is clinically significant. The application of these thresholds and the integration of the ImRomberg test with the clinical tandem gait test allowed for the detection of balance impairments in 58% of early-stage PwMS versus 17% disclosed by administering the traditional clinical Romberg test and tandem gait test. The significantly higher sensitivity of the proposed approach with minimal modifications to the tests commonly performed in clinical practice would allow for the identification of PwMS (who should begin an early rehabilitation training) in the very early stages of the disease and directly during neurological examination.

## Figures and Tables

**Figure 1 sensors-22-09558-f001:**
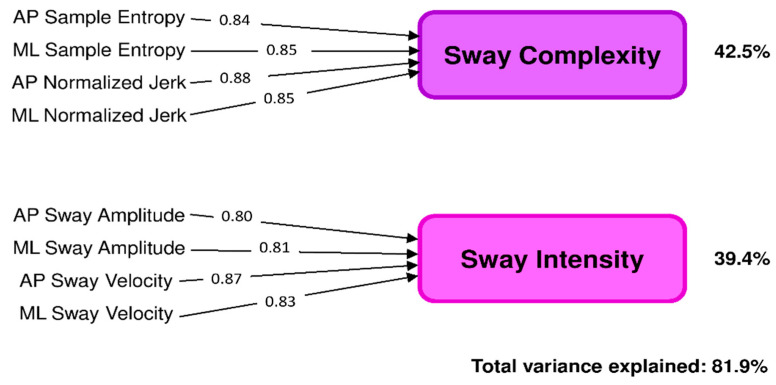
Result of the principal component analysis. Eight wearable-sensor-based variables descriptive of trunk sway during the instrumented modified Romberg test (reported on the left) were reduced to two “latent variables” (principal components) called “sway complexity” and “sway intensity”. Arrows report significant factor loadings (>0.60). The percentage of variance explained by each component and the total variance is given on the right.

**Figure 2 sensors-22-09558-f002:**
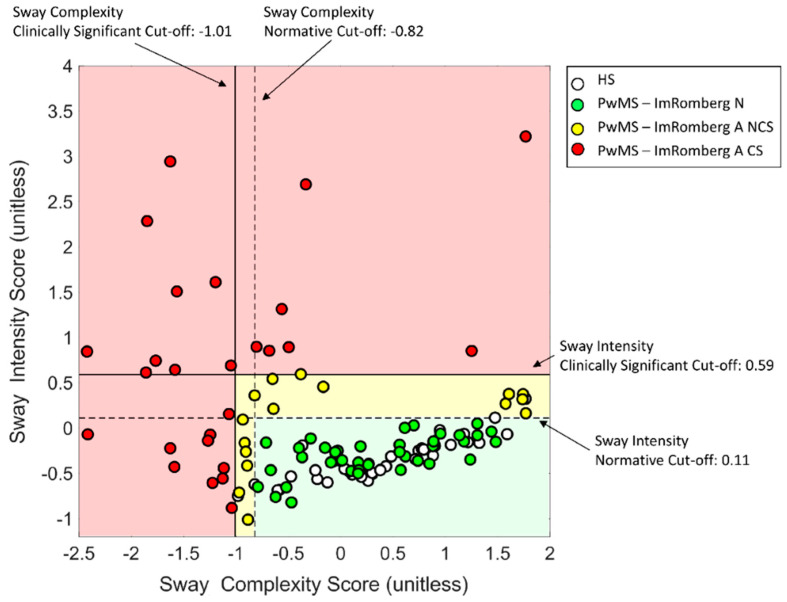
Scatterplot reporting sway complexity and sway intensity scores extracted from the ImRomberg in each participant. HS: healthy subjects; PwMS: people with multiple sclerosis; N: normal; A NCS: abnormal, not clinically significant, A CS: abnormal, clinically significant.

**Figure 3 sensors-22-09558-f003:**
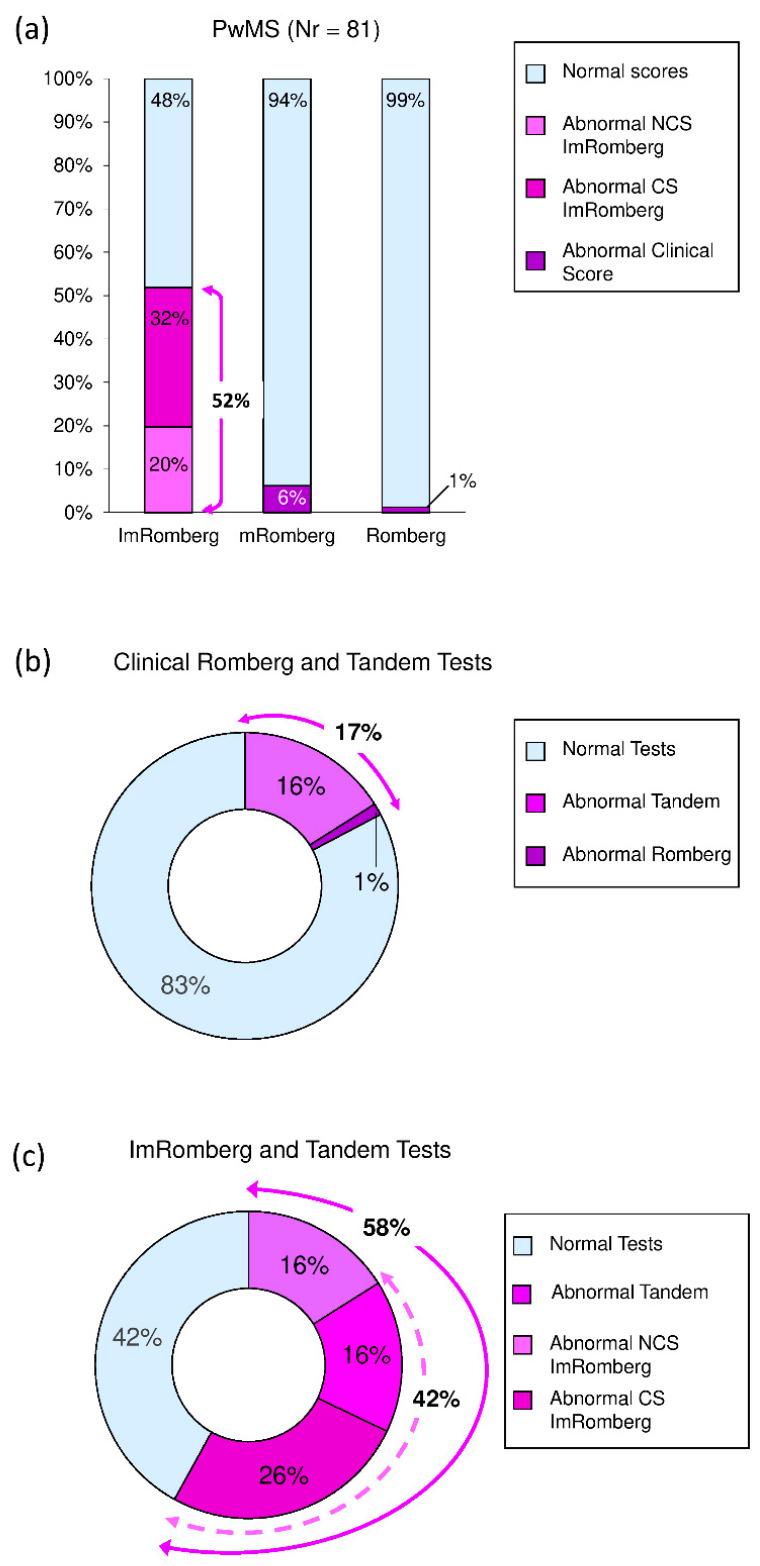
(**a**) Percentage of people with MS (PwMS) showing abnormal ImRomberg test and abnormal clinical scores on modified Romberg (mRomberg) and Romberg tests. (**b**) Percentage of PwMS with abnormal tandem gait test (light violet) and with normal tandem but abnormal clinical Romberg test (dark violet). (**c**) Percentage of PwMS with abnormal tandem gait test (light violet) and with normal tandem but abnormal ImRomberg (light and dark magenta). ImRomberg was considered abnormal in case of sway complexity and/or sway intensity score outside normative cut-offs (two-headed dashed arrow). Abnormal NCS: abnormal, not clinically significant (sway complexity and/or sway intensity outside normative cut-offs but inside clinically significant cut-offs); abnormal CS: abnormal, clinically significant (sway complexity and/or sway intensity outside clinically significant cut-offs).

**Figure 4 sensors-22-09558-f004:**
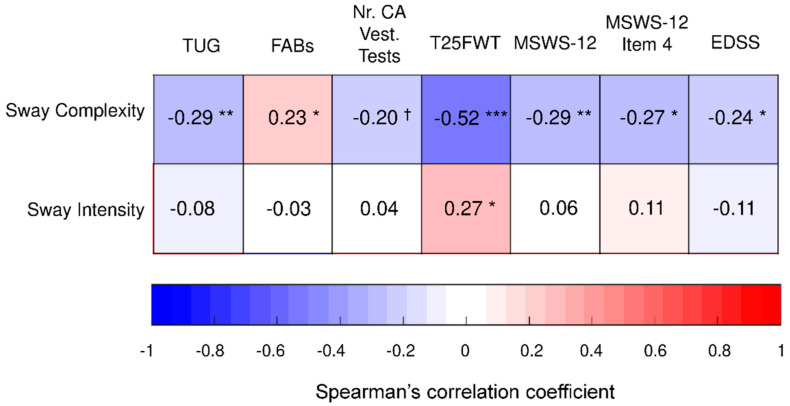
Spearman’s correlation coefficient between ImRomberg components (sway complexity and sway intensity) and clinical measures in PwMS. *** *p* < 0.001, ** *p* < 0.01; * *p* < 0.05; ^†^
*p* < 0.10. TUG: timed up and go; FABs: Fullerton advanced balance scale—short; Nr. CA Vest. Tests: number of clinically abnormal vestibular tests (i.e., standing on foam with eyes closed, turning 360° left and right, walking with head turns); T25FWT: timed 25 foot walk test; MSWS-12: twelve-item multiple sclerosis walking scale; EDSS: expanded disability status scale.

**Table 1 sensors-22-09558-t001:** Description of the wearable-sensor-based variables computed during the instrumented modified Romberg Test (ImRomberg).

Variable	Description
AP (ML) Sway Amplitude (SwAmp) [m/s^2^]	Root mean square of AP (ML) acceleration signal [7].
AP (ML) Sway Range (SwRange) [m/s^2^]	Range (maximum−minimum value) of AP (ML) acceleration signal [46].
95% Confidence Ellipse Area (SwArea) [m^2^/s^4^]	Area of the ellipse containing 95% of ML/AP acceleration data points computed following Schubert and Kirchner [51]
AP (ML) Sway Velocity (SwVel) [m/s]	Mean of the absolute value of the AP (ML) velocity signal obtained by integrating the AP (ML) acceleration. Before integration, the accelerations were high-pass filtered at 0.15 Hz with a zero-lag, fourth order Butterworth filter to limit the drift effect [52].
AP (ML) Sway Path (SwPath) [m]	Mean length of the AP (ML) trajectory traveled by the trunk and calculated as the product between AP (ML) sway velocity and the duration of the test [53].
AP (ML) Normalized Jerk (nJerk) [-]	Logarithm of the normalized AP (ML) jerk (i.e., first time derivative of the acceleration) computed as described by Caby et al. [54]. In particular, AP (ML) jerk was normalized with respect to the range of AP (ML) acceleration and the test duration (see [7]). Decreasing values of nJerk indicate smoother trunk sway.
AP (ML) Total Spectral Power (Pwr) [m^2^/s^4^]	Integrated area of the power spectral density of AP (ML) acceleration computed using the Welch method with a Hanning window of 5 s and 50% overlap [52]. Increasing values of Pwr indicate higher energy expenditure [53]
AP (ML) 95% Power Frequency (F95) [Hz]	Frequency below which 95% of the AP (ML) acceleration power is contained [55]. Higher values of F95 indicate a higher frequency of trunk sway.
AP (ML) Centroidal Frequency (CF) [Hz]	Frequency at which the spectral mass of AP (ML) acceleration is concentrated [7,53]
AP (ML) Frequency Dispersion (FD) [-]	Measure of the variability of frequency content of the power spectral density (zero for pure sinusoid, increases with spectral bandwidth to one) [46,53]
AP (ML) Sample Entropy (SaEn) [-]	SaEn was computed on the standardized AP (ML) acceleration obtained by subtracting the mean and dividing by the standard deviation of the signal. SaEn is related to the conditional probability that two sequences of *m* consecutive data points similar to each other (i.e., distance between data points lower than a tolerance *r*) will remain similar when one more consecutive point is included [56,57]. Values of *m* and *r* were set equal to 2 and 0.15, respectively, following Busa et al. [47]. SaEn is a measure of the regularity and predictability of AP (ML) trunk accelerations, thus providing information on the complexity of the signal [49,57]. The higher SaEn, the more complex (less regular) the trunk acceleration.

AP: anteroposterior; ML: mediolateral.

**Table 2 sensors-22-09558-t002:** Demographic and clinical characteristics of healthy subjects (HS) and people with multiple sclerosis (PwMS).

Variable	HS (*N* = 38)	PwMS (*N* = 81)	*p*-Value
Age [years]	34 (24; 58)	39 (25; 56)	0.360
Female [*N* (%)]	22 (58%)	53 (65%)	0.427
Disease Duration [years]	-	2 (0; 5)	-
EDSS Score [0–10]	-	1.5 (0; 2.5)	-
MS Type [*N* (%)]			
Relapsing-Remitting	-	80 (99%)	-
Primary Progressive	-	1 (1%)	-
Secondary Progressive	-	0 (0%)	-
Abnormal Romberg Test [*N* (%)]	0 (0%)	1 (1.2%)	0.555
Abnormal Tandem Test [*N* (%)]	0 (0%)	13 (16%)	0.024
Abnormal mRomberg Test [*N* (%)]	0 (0%)	5 (6%)	0.150
FABs [0–24]	24 (24; 24)	23 (18; 24)	<0.001
TUG [s]	-	7.3 (5.2; 9.7)	-
T25FWT [s]	-	4.0 (3.2; 5.7)	-
MSWS-12 score [0–100]	-	4.2 (0; 43.8)	-
MSWS-12 Item 4 [1–5]	-	1 (1; 3)	-

Values are median (5th; 95th percentile) or number (percentage). *p*-value refers to Mann–Whitney U Test for age and FABs and chi-square test for all the other variables. EDSS: expanded disability status scale; mRomberg: modified Romberg test; FABs: Fullerton advanced balance scale—short; TUG: timed up and go test; T25FWT: timed 25 foot walk test; MSWS-12: twelve-item multiple sclerosis walking scale.

**Table 3 sensors-22-09558-t003:** Wearable-sensor-based variables were computed during the instrumented modified Romberg test (ImRomberg) executed by healthy subjects (HS) and people with multiple sclerosis (PwMS).

Variable	HS (*N* = 38)	PwMS (*N* = 81)	*p*-Value	∆%
AP Sway Amplitude [m/s^2^]	0.12 (0.08; 0.18)	0.17 (0.09; 0.58)	<0.001	+42%
ML Sway Amplitude [m/s^2^]	0.07 (0.04; 0.13)	0.10 (0.05; 0.57)	<0.001	+43%
AP Sway Range [m/s^2^]	0.78 (0.51; 1.20)	1.03 (0.54; 5.54)	<0.001	+32%
ML Sway Range [m/s^2^]	0.43 (0.25; 0.87)	0.63 (0.32; 4.98)	<0.001	+47%
95% Conf. Ellipse Area [m^2^/s^4^]	0.15 (0.08; 0.35)	0.24 (0.10; 6.19)	<0.001	+60%
AP Sway Velocity [m/s]	0.05 (0.02; 0.09)	0.08 (0.03; 0.30)	<0.001	+60%
ML Sway Velocity [m/s]	0.02 (0.01; 0.08)	0.05 (0.02; 0.33)	<0.001	+150%
AP Sway Path [m]	1.07 (0.42; 1.74)	1.64 (0.58; 582)	<0.001	+53%
ML Sway Path [m]	0.49 (0.22; 1.56)	1.16 (0.31; 6.58)	<0.001	+137%
AP Normalized Jerk [-]	3.74 (3.27; 4.45)	3.53 (2.89; 4.32)	0.007	−6%
ML Normalized Jerk [-]	3.75 (3.19; 4.28)	3.55 (2.92; 4.10)	0.002	−5%
AP Total Spectral Power [m^2^/s^4^]	0.08 (0.04; 0.16)	0.11 (0.04; 1.79)	0.002	+38%
ML Total Spectral Power [m^2^/s^4^]	0.03 (0.01; 0.11)	0.06 (0.01; 1.25)	<0.001	+100%
AP 95% power frequency [Hz]	1.61 (1.03; 2.34)	1.76 (0.88; 2.78)	0.281	+9%
ML 95% power frequency [Hz]	1.83 (1.17; 2.93)	1.90 (1.03; 2.93)	0.620	+4%
AP Centroidal Frequency [Hz]	0.76 (0.56; 1.02)	0.81 (0.53; 1.23)	0.243	+7%
ML Centroidal Frequency [Hz]	0.91 (0.62; 1.16)	0.89 (0.58; 1.37)	0.750	−2%
AP Frequency Dispersion [-]	0.68 (0.58; 0.76)	0.66 (0.54; 0.75)	0.153	−9%
ML Frequency Dispersion [-]	0.63 (0.52; 0.72)	0.62 (0.50; 0.72)	0.869	−2%
AP Sample Entropy [-]	1.73 (1.12; 2.13)	1.34 (0.53; 2.21)	0.010	−23%
ML Sample Entropy [-]	1.77 (1.16; 2.23)	1.33 (0.46; 2.09)	<0.001	−25%

Values are median (5th; 95th percentile). *p*-value refers to Mann–Whitney U Test with Benjamini–Hochberg correction for multiple comparisons. AP: anteroposterior; ML: mediolateral. ∆%: (median PwMS−median HS)/median HS.

**Table 4 sensors-22-09558-t004:** ImRomberg components scores for healthy subjects (HS) and people with multiple sclerosis (PwMS).

Variable	HS (*N* = 38)	PwMS (*N* = 81)	*p*-Value	∆%
Sway Complexity [-]	0.29 (−0.82; 1.59)	−0.37 (−1.77; 1.61)	0.003	−228%
Sway Intensity [-]	−0.41 (−0.69; 0.11)	−0.15 (−0.72; 1.61)	<0.001	+63%

Values are median (5th; 95th percentile). *p*-value refers to Mann–Whitney U Test with Benjamini–Hochberg correction for multiple comparisons. ∆%: (median PwMS−median HS)/median HS.

**Table 5 sensors-22-09558-t005:** ImRomberg components scores for participants with and without clinically normal balance.

Variable	Clinically Normal Balance Group (*N* = 63)	Clinically Abnormal Balance Group (*N* = 56)	*p*-Value	∆%
Sway Complexity [-]	0.27 (−1.01; 1.59)	−0.48 (−1.18; 1.74)	0.002	−278%
Sway Intensity [-]	−0.31 (−0.63; 0.59)	−0.15 (−0.83; 2.69)	0.041	+52%

Values are median (5th; 95th percentile). *p*-value refers to Mann–Whitney U Test with Benjamini–Hochberg correction for multiple comparisons. ∆%: (median clinically abnormal balance group−median clinically normal balance group)/median clinically normal balance group.

## Data Availability

The dataset used and/or analyzed during the current study are available from the corresponding author under reasonable request.

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
