# Peer review of "Balance Impairments in People with Early-Stage Multiple Sclerosis: Boosting the Integration of Instrumented Assessment in Clinical Practice"

_sensors, 2022, doi:10.3390/s22239558_

Round 1

Reviewer 1 Report

In the submitted manuscript, the authors describe their study in which they utilize an instrumented modified Romberg test (standing on foam with eyes closed, wearing an IMU on the trunk) to quantify the balance of a large group of healthy subjects and a large group of early-stage Multiple Sclerosis subjects. From the IMU data captured during the balance test, the authors calculated 21 IMU-based metrics, many of which were significantly different between the two groups. In addition, they utilize principal component analysis to reduce the metrics. The first two components captured over 80% of the variability in the dataset and were defined as Sway Complexity and Sway Intensity. The authors demonstrate that the instrumented test is more sensitive to balance abnormalities than the standard clinical tests, and also use their dataset to define normative ranges so that clinicians can make informed decisions and identify patients who could benefit from preventative rehabilitation.

Overall, I feel this is a significant contribution to the literature. I am currently in the middle of a study in which we are also using IMUs to create a more sensitive balance assessment, and I really enjoyed reading this manuscript. The paper is very well written, has great figures, and presents the results in a very clear manner. I was very impressed that the authors provide Table B1, which gives the values needed to calculate Sway Complexity and Sway Intensity from the more traditional IMU-based balance metrics; I read many articles in which the authors use machine learning but do not provide the model, so this paper was a refreshing contrast.

I have no suggestions to the authors other than to review/proofread the article again, as there are only minor changes needed in the text (none are significant enough to list here).

Author Response

Point 1. In the submitted manuscript, the authors describe their study in which they utilize an instrumented modified Romberg test (standing on foam with eyes closed, wearing an IMU on the trunk) to quantify the balance of a large group of healthy subjects and a large group of early-stage Multiple Sclerosis subjects. From the IMU data captured during the balance test, the authors calculated 21 IMU-based metrics, many of which were significantly different between the two groups. In addition, they utilize principal component analysis to reduce the metrics. The first two components captured over 80% of the variability in the dataset and were defined as Sway Complexity and Sway Intensity. The authors demonstrate that the instrumented test is more sensitive to balance abnormalities than the standard clinical tests, and also use their dataset to define normative ranges so that clinicians can make informed decisions and identify patients who could benefit from preventative rehabilitation.

Overall, I feel this is a significant contribution to the literature. I am currently in the middle of a study in which we are also using IMUs to create a more sensitive balance assessment, and I really enjoyed reading this manuscript. The paper is very well written, has great figures, and presents the results in a very clear manner. I was very impressed that the authors provide Table B1, which gives the values needed to calculate Sway Complexity and Sway Intensity from the more traditional IMU-based balance metrics; I read many articles in which the authors use machine learning but do not provide the model, so this paper was a refreshing contrast.

Response 1. We really thank the reviewer for his/her appreciation about all mentioned points.

Point 2. I have no suggestions to the authors other than to review/proofread the article again, as there are only minor changes needed in the text (none are significant enough to list here).

Response 2. We thank the reviewer for the suggestion. The manuscript has been proofread.

Reviewer 2 Report

Below are my comments

Introduction

Great job writing the introduction and suggesting why there is a need to use the modified Romberg

Methodology

The methodology needs a bit more clarification

1. How did you determine that a sample size of 38 was sufficient to compare your healthy population to PwMS?

2. Did you match the participants?

3. I really appreciate the description of the 21 variables computed for this study however, my suggestion would be to put them in a table as it would make it significantly easier for the reader

4. What was the order in which the tests were performed? Were they performed on a single day? I know that you have already published this work, but it would help for the readers of this study to understand when the tests were performed.

5. Did the participants wear the IMU sensors throughout the duration of the tests and only data from the FABs were utilized? 

6. The statistical analyses section in my opinion is the strongest part of this paper. The explanations are very clear and very helpful to the reader. 

7. Did you measure height and weight? Were there significant differences? There is significant evidence that weight impacts balance control. If not, please report that in your limitations section.

Results

1. I don't know if I'd consider 0.68 and 0.72 acceptable for AUC mean. Most models with AUC <0.80 have significant errors. However, if the authors can provide a citation or justification for why this interpretation is appropriate I think the authors should provide it.

2. Figure 2 was extremely useful

3. Same with Figure 3

Discussion

The discussion is very well written and does a great job of explaining the results. 

If weight data was not collected, please make sure you reference that in your limitations.

Some other thoughts

Have you thought about using classifier models to predict HS vs PwMS using either DL or ML with your data? Based on the results you have presented it seems that you might be able to get strong models with good classification accuracy and precision. 

Round 2

Reviewer 2 Report

I'd like to thank the authors for addressing all of my queries.